# Development and Optimal Immune Strategy of an Alum-Stabilized *Pickering emulsion* for Cancer Vaccines

**DOI:** 10.3390/vaccines11071169

**Published:** 2023-06-28

**Authors:** Sha Peng, Yumeng Yan, To Ngai, Jianjun Li, Kenji Ogino, Yufei Xia

**Affiliations:** 1State Key Laboratory of Biochemical Engineering, Institute of Process Engineering, Chinese Academy of Sciences, Beijing 100190, Chinayanyumeng21@ipe.ac.cn (Y.Y.); jjli@ipe.ac.cn (J.L.); 2Graduate School of Bio-Applications and Systems Engineering, Tokyo University of Agriculture and Technology, 2-24-16 Nakacho, Tokyo 184-8588, Japan; 3Institute of Global Innovation Research, Tokyo University of Agriculture and Technology, 2-24-16 Nakacho, Tokyo 184-8588, Japan; 4University of Chinese Academy of Sciences, Beijing 100049, China; 5Department of Chemistry, The Chinese University of Hong Kong, Shatin, N.T., Hong Kong, China; tongai@cuhk.edu.hk; 6Innovation Academy for Green Manufacture, Chinese Academy of Sciences, Beijing 100190, China

**Keywords:** cancer vaccine, alum-stabilized *Pickering emulsion*, adjuvant, anti-tumor responses

## Abstract

Therapeutic cancer vaccines are considered as one of the most cost-effective ways to eliminate cancer cells. Although many efforts have been invested into improving their therapeutic effect, transient maturation and activations of dendritic cells (DCs) cause weak responses and hamper the subsequent T cell responses. Here, we report on an alum-stabilized *Pickering emulsion* (APE) that can load a high number of antigens and continue to release them for extensive maturation and activations of antigen-presenting cells (APCs). After two vaccinations, APE/OVA induced both IFN-γ-secreting T cells (Th1) and IL-4-secreting T cells (Th2), generating effector CD8^+^ T cells against tumor growth. Additionally, although they boosted the cellular immune responses in the spleen, we found that multiple administrations of cancer vaccines (three or four times in 3-day intervals) may increase the immunosuppression with more PD-1^+^ CD8^+^ and LAG-3^+^ CD8^+^ T cells within the tumor environment, leading to the diminished overall anti-tumor efficacy. Combining this with anti-PD-1 antibodies evidently hindered the suppressive effect of multiple vaccine administrations, leading to the amplified tumor regression in B16-OVA-bearing mice.

## 1. Introduction

As a safe and effective therapeutic modality, cancer immunotherapy aims to activate the endogenous immune responsiveness and eliminate malignant cells by eliciting antigen-directed cytotoxicity of functional tumor-specific cytotoxic T cells [1,2,3]. Cancer immunotherapy approaches are classified into several categories, and tumor vaccination, as one of the most attractive immunotherapy strategies, provokes a persistent T cell activation and acquisition of effector functions, and it displays clear rationale and compelling preclinical data for further development [4,5]. In order to elicit effector T cell for recognition and elimination of tumor cells, the ideal vaccines are expected to deliver antigens to the antigen-presenting cells (APCs), such as dendritic cells (DCs), and allow them to activate with immunogenic signals and process and present the antigens [6,7]. In this step, fluid antigens fail to potently enter the APCs, and there is a demand for delivery vehicles or adjuvants to stimulate the activations of APCs [8]. Next, the primed DCs, with a high expression of activation and/or maturation signals, would further differentiate and migrate to the lymph nodes to present antigens to naive T cells [9,10], with the robust secretion of a series of immunogenic cytokines, such as IL-12, IFN-α, IL-10, etc. [11]. The activations of APCs should lead to the expansion of T cells in sufficient amounts for recognizing and eliminating malignant cells [12]. Accordingly, dynamic APC activations, which capture, process, and present tumor antigens to T cells systematically and act as immunostimulants, are vital for the onset of anti-tumor effects in cancer vaccines [13].

Based on our previous research, with the ability to efficiently load antigens and enter APCs, and elicit robust and durable activation, particle-stabilized emulsion (*Pickering emulsion*) composed of oil and nanoparticles is regarded as a potential vaccine candidate for cancer therapies [14,15]. Furthermore, alum-stabilized *Pickering emulsion* (APE) using a commercialized aluminum adjuvant (termed “alums”) and the US Food and Drug Administration (FDA)-approved squalene is regarded as able to achieve safe transfer in the clinic [16]. Simply preparing and long-term preserving of it further promote its clinical translation. Moreover, APE was developed with the following functions. (1) The particle-adsorbed interphase offers higher adsorption of antigens, leading to amplified antigen delivery. (2) Hydrophobicity of the APE allows for increased APC uptake and activations. (3) The delivered antigens can escape from lysosomes by APE because of the positive charge of the oil/water interface, facilitating the cross-presentation to T cells for higher cellular immune responses. The abovementioned advantages have led to a mixed Th1 and Th2 activation by APE, which was found to offer comprehensive immune protection against COVID-19 [17]. To broaden its clinical applicability, the potential of APE as a tumor vaccine should be investigated. To achieve the optimal therapeutic effect, it is necessary to explore the best immunization strategy, including the use of booster vaccinations. Increasing the number of administrations is frequently employed in clinical tumor vaccines to increase anti-tumor efficacy. However, in many trials, the effectiveness remains limited after frequent vaccinations and with advancing cancer growth [18]. Numerous studies have shown that persistent antigen presentation could lead to amplification of exhausted T cells, thus resulting in a decrease in the effector functions and an increase in the expression of PD-1 in the immunosuppressive microenvironment for the hindered tumor regression [19,20,21]. Therefore, developing an optimal immune strategy of APE for cancer vaccines is crucial to enhance the anti-tumor efficacy.

Here, we evaluated the potential of alum-stabilized *Pickering emulsion* as a cancer vaccine and optimized the immune strategy. Chicken egg albumin (OVA) [22], composed of 386 amino acids and with a molecular weight of about 45 kD, was selected as the model antigen. We first proved the T cell engagement after intramuscular injection of APE/OVA. Additionally, the effectiveness against tumor regression of two administrations (APE-2), three administrations (APE-3), and four administrations (APE-4) of the vaccine was estimated and compared. A more detailed analysis in tumor draining lymph nodes (TDLNs) and the tumor microenvironment (TME) was undertaken to assess the underlying mode of action of the anti-tumor effect of multiple vaccinations. Finally, we combined the APE-adjuvant vaccine with anti-programmed cell death 1 ligand 1 antibody (anti-PD-1) to further amplify the effector T cell potency and induce potent anti-tumor immune responses.

## 2. Materials and Methods

### 2.1. Materials

Alum Hydroxide Gel Adjuvant (10 mg/mL) was purchased from invivoGen (San Diego, CA, USA). The micro-BCA kit, ovalbumin (OVA), and FDA-approved squalene were purchased from Sigma (Saint Louis, MI, USA). Penicillin–streptomycin, Dulbecco’s modified Eagle medium (DMEM), Roswell Park Memorial Institute (RPMI) 1640, and fetal bovine serum (FBS) were purchased from GIBCO BRL (Gaithersburg, MD, USA). Mouse IFN-γ and IL-4 ELISpot BASIC kits and BCIP/NBT-plus substrate for ELISpot were purchased from Mabtech (Stockholm, Sweden). Poly vinylidene difluoride-backed plates were received from Merck-Millipore (Burlington, MA, USA). All other reagents used in this study were of analytical grade.

### 2.2. Cell Culture

B16-OVA tumor cells were obtained from ATCC (Manassas, VA, USA) and were cultured in DMEM. The cell lines used in this study were confirmed to be without mycoplasma contamination. All media included both 10% heat-inactivated FBS (*v*/*v*) and 1% penicillin–streptomycin (*v/v*).

### 2.3. Mice

C57BL/6 mice (6- to 8-week-old females) were received from Vital River Animal Laboratories (Beijing, China). All mice were housed in ventilated cages without pathogens and offered suitable humidity, temperature, and light–dark cycles. All animal experiments were reviewed and approved by the Institutional Animal Use Committee of the Institute of Process Engineering (Chinese Academy of Sciences).

### 2.4. Preparation of APE

APE was prepared through a single-step sonication of alum as a colloidal stabilizer and squalene as a dispersion phase. Specifically, alum was dispersed in ultrapure water to form the aqueous phase. Squalene, which served as the oil phase, was further added at an oil/water ratio of 1/9 (*v*/*v*). Then, uniform emulsion droplets were prepared by single-step sonication (Branson Digital Sonifier, total time = 120 s, power = 30%, interval time = 4 s). OVA as a model antigen was assembled on the emulsion droplets through electrostatic adsorption, and the loading efficiency was measured by a micro-BCA kit.

### 2.5. Vaccination Study

Female C57BL/6 mice (6–8 weeks old) (*n* = 6 mice per group) were intramuscularly immunized with 100 μL APE (100 μg alum + 5 μL squalene) containing 10 μg OVA per mouse based on the indicated schedule (Figure 1). One week after the final immunization, the mice were sacrificed and their spleens were collected for subsequent immune assessment.

### 2.6. ELISpot Evaluations

For ELISpot evaluations, 96-well PVDF-packed plates (Millipore) were activated by 35% ethyl alcohol followed by washing 5 times with pure water. Then, the plates were coated with anti-mouse IFN-γ or IL-4 antibodies (5 μg/mL, 100 µL/well) overnight at 4 °C. The next day, after washing two times, the plates were blocked using culture medium (RPMI 1640 with 10% FBS) in the dark for at least 1 h. Then, splenocytes from vaccinated mice 7 days after the final administration were added (5 × 10^5^ cells/well, 100 µL/well) with OVA model antigen (2 μg/mL) and cultured for 24 h (37 °C, 5% CO_2_) without moving the plates. After removing cells and washing five times, the plates were incubated in the dark using detection antibody (1 μg/mL, 100 µL/well) for more than 2 h, followed by streptavidin-ALP detection antibody (1 μg/mL, 100 µL/well) at 25 °C. Then, the BCIP/NBT (100 µL/well) was added to the plates and incubated for about 5 min without light. Finally, the plates were washed with pure water and air-dried without light overnight. The spots were scanned and counted via an ELISpot Analyzer (AT-Spot 2100, Antai Yongxin Medical Technology, Beijing, China). Each of the experiments was repeated three times, obtaining similar results each time.

### 2.7. Cytokine Secretion and Immune Cells in the Spleen

The vaccinated mice were sacrificed humanely 7 days after the final administration, and the spleens were isolated for further mechanical disruption for the preparation of cell suspensions. After lysis of the red blood cells, the single-cell suspensions were added to 24-well plates (2 × 10^6^ cells/well) and restimulated with the OVA (2 μg/mL) at 37 °C, 5% CO_2_ for 48 h. Next, the supernatant was collected by centrifugation (10,000× *g*, 5 min) for further cytokine secretion assays (IL-2, IL-4, IFN-γ, TNF-α, granular enzyme B) via ELISA kits from Solarbio (Beijing, China). The treated cells were collected (500× *g*, 5 min) and stained with fluorescent antibodies stained with Ghost Dye™ Violet 450, APC-Cy7 anti-mouse CD3e antibody, PerCP-Cyanine5.5 anti-mouse CD8 antibody, and PE anti-mouse SIINFEKL-Pentamer antibody and analyzed by flow cytometry. According to the manufacturer’s handbook, the antibody dilution was performed for flow cytometry staining. Data were collected by flow cytometry (CytoFlex LX, Beckman Coulter, Brea, CA, USA). Three repetitions of each experiment were carried out.

### 2.8. Cytokine Secretion and Immune Cells in Tumors and TDLNs

For the tumor-bearing mice, 3 days after the last immunization, the mice were sacrificed humanely and the tumors and TDLNs were isolated for further mechanical disruption for the preparation of single-cell suspensions. Specifically, TDLNs were mechanically disrupted and resuspended in PBS. Single-cell suspensions obtained from the TDLNs were collected by centrifugation (500× *g*, 5 min) and stained with fluorescent antibodies stained with Ghost Dye™ Violet 450, APC-Cy7 anti-mouse CD3e antibody, PerCP-Cyanine5.5 anti-mouse CD8 antibody, APC anti-mouse LAG-3 antibody, and FITC anti-mouse PD-1 antibody.

For immune cell analysis in tumors, the tumors were suspended in whole blood and tissue diluent (100 mg tissue/mL) and mechanically disrupted. Tumor-infiltrating lymphocytes (TILs) were obtained by using a mouse tumor-infiltrating tissue lymphocyte isolation kit (Solarbio, Beijing, China), following the manufacturer’s instructions. Firstly, the single cells were collected at 500× *g* for 5 min at 4 °C and resuspended in 5 mL of whole blood and tissue diluent. Next, the single-cell suspension was slowly dropped into 5 mL of lymphocyte isolation fluid for tumor-infiltrating tissue and centrifuged at 900× *g* for 30 min at 4 °C. Finally, lymphocytes were gently collected, washed three times using washing solution, and stained with fluorescent antibodies stained with Ghost Dye™ Violet 450, APC-Cy7 anti-mouse CD3e antibody, PerCP-Cyanine5.5 anti-mouse CD8 antibody, APC anti-mouse LAG-3 antibody, and FITC anti-mouse PD-1 antibody, before being analyzed by flow cytometry.

According to the manufacturer’s handbook, the antibody dilution was performed for flow cytometry staining. Data were collected by flow cytometry (CytoFlex LX, Beckman Coulter, Brea, CA, USA). Three repetitions of each experiment were carried out.

### 2.9. Evaluation of Health

For detection of the serum biochemical parameters, 7 days after final vaccination, blood supernatant samples were collected through the retro-orbital route, and we analyzed the levels of AST, ALT, BUN, ALP, and LDH via an automated analyzer (Hitachi 917, Hitachi Ltd., Tokyo, Japan). To determine the morphological changes in the main visceral organs, the hearts, spleens, livers, kidneys, and lungs were harvested and fixed in 4% (*v*/*v*) formaldehyde and sectioned for H&E staining.

### 2.10. Evaluation of Tumor Growth Inhibition Using a B16-OVA-Bearing Mouse Model

Single B16-OVA tumor cells (5 × 10^5^ cells per mouse) were subcutaneously inoculated into the left axillary region of C57BL/6 mice (6-week-old females). APE (100 μg alum + 5 μL squalene) with OVA (10 μg) was injected based on the indicated schedule after random assigning of tumor-bearing mice to different groups (eight mice per group). Mice of the control group were PBS-treated simultaneously. To monitor tumor progression, tumor growth and survival of the mice were measured every two days. The volume of the tumor was calculated using the following formula: (long-axis diameter) × (short-axis diameter)^2^/2.

### 2.11. Combination with Anti-PD-1 Antibody Using a B16-OVA-Bearing Mouse Model

Six days after of tumor inoculation (5 × 10^5^ cells per mouse), the mice received APE (100 μg alum + 5 μL squalene), with OVA (10 μg), according to the indicated schedule, after random assigning of tumor-bearing mice to different groups (eight mice per group). Anti-PD-1 antibody (clone: RMP1-14, BioXCell, 100 μg) immunizations were performed every 3 days for a total of three times based on the indicated schedule.

### 2.12. Statistical Analysis

All animal studies were performed after randomization. All values were expressed as means ± s.e.m. Data were analyzed by one- or two-way analysis of variance (ANOVA) for comparison of multiple groups with GraphPad Prism 9. Flow cytometry data were analyzed using FlowJo 7.6 and CytExpert. *p* values less than 0.05 were considered statistically significant.

## 3. Results

### 3.1. Alum-Stabilized Pickering Emulsion Preparation and Characterization

We prepared alum-stabilized *Pickering emulsion* using a previously reported method [23]. After conducting a series of optimizations, alum hydrogel (2 mg/mL), which is a commercially employed adjuvant, was dispersed in sterile deionized water (pH 7) as the aqueous phase, then mixed with the oil phase (squalene) to form the emulsion droplets. The size and zeta potentials of APE and the corresponding alum hydrogel (alum) are shown in Table 1.

In this study, ovalbumin (OVA) was selected as the model antigen, which is recognized by the T cell-mediated immune system and activates the T cell immune response [24]. The particle-adsorbed interphase of APE offered a large specific surface area that facilitated the adsorption of OVA. Moreover, the positively charged surface further increased the electrostatic adsorption of negatively charged OVA. Thus, APE adsorbed more than 90% of the OVA antigen (10 µg/per mouse) within 1 h (Table 1), indicating that it was equipped to effectively trigger an immune response.

### 3.2. Adjuvant Effects of Alum-Stabilized Pickering Emulsion as a Tumor Vaccination

We postulated that APE could modulate the adaptive immune responses in tumor vaccinations. To test our hypothesis, the adjuvanticity of APE/OVA formulations, which contained 100 µg of alum and 10 µg of OVA antigen per mouse, was investigated. Compared with the control group, in the spleen, over 15-fold more OVA-specific IFN-γ-secreting T cells (Th1 cells) were detected after prime-post intramuscular injection of APE/OVA (APE-2), as illustrated in Figure 2a. Furthermore, increasing the administration frequency also further boosted the IFN-γ-mediated T cell responses. As shown in Figure 2b, 2-fold and 3-fold more IFN-γ-secreting T cells were detected after three (APE-3) and four (APE-4) intramuscular injections, compared with two intramuscular injections (APE-2). The amount of IL-4-secreting cells (Th2 cells) was also 5-fold higher than the control group after two intramuscular injections, and 2-fold and 3-fold more IL-4 spot-forming cells were also detected in APE-3 and APE-4, compared with APE-2 (Figure 2c,d).

Moreover, the IL-4 and IFN-γ cytokine profile of the splenocyte supernatant showed a similar tendency. In comparison to the control group, APE-2 increased IFN-γ secretion by 1260% (Figure 3a) and IL-4 secretion by 200% (Figure 3b). With the increase in the frequency of vaccinations, more robust secretion of IFN-γ (~860%) and IL-4 (~370%) was detected in APE-4 compared with APE-2 (Figure 3).

Next, the more specific cytokine profile, which was related to tumor immunity, was assessed in the splenocytes. APE triggered significantly enhanced cytokine secretion levels compared with the control. In development, APE-4 increased the concentration of IL-2 secretion by 140% (Figure 4a), TNF-α secretion by 70% (Figure 4b), and granzyme B secretion (Figure 4c) by 230% compared with APE-2-treated mice, indicating potentially robust cellular immune engagement against tumors. As shown in Figure 4d,e, in the spleen of APE-adjuvant mice, the frequency of OVA-specific CD8^+^ T cells was increased by about 100% after receiving two intramuscular vaccines, and they were further stimulated by 160% after four-dose vaccinations, compared with non-treated mice (control), as shown in Appendix A and Figure 4d,e. These results suggested that APE may harbor the potential to elicit a robust anti-tumor effect. Moreover, boosting administrations of APE (APE-3 and APE-4) may offer a higher cellular immune response.

### 3.3. The Safety Profile of APE after Multiple Immunizations

The safety profile of APE after four immunizations (APE-4) was evaluated by histopathology analysis of the major organs and serum biochemical parameters of the key factors. No noticeable signs of side effects were observed in the heart, liver, spleen, lung, and kidney after four administrations (Figure 5), compared with the non-treated group (control). Additionally, the levels of lactate dehydrogenase (LDH), aspartate aminotransferase (AST), alanine aminotransferase (ALT), alkaline phosphatase (ALP), and blood urea nitrogen (BUN) were within a similar range to those of the PBS-treated mice (control), as shown in Appendix A.

### 3.4. Anti-Tumor Therapeutic Efficacy

The above evaluations encouraged us to conduct a further appraisal of the therapeutic effect in an established B16-OVA cancer model. The C57BL/6 mice were implanted subcutaneously with B16-OVA cells (5 × 10^5^/per mouse) on D + 0 (Day 0), and subsequently received the vaccinations of APE at various times (Figure 6a, each dose containing 100 µg of alum and 10 µg of OVA). Intramuscular administration with APE/OVA delayed the onset of the B16-OVA tumor model (Figure 6b,c) and increased the mouse survival rates (Figure 6d), with a stable body weight (Appendix A). These results demonstrated that APE functioned as a potent adjuvant to dampen the activity of tumor proliferation. Surprisingly, unlike the results of the cellular immune responses, the intensified tumor regression was not observed after further immunizations (APE-3 and APE-4).

### 3.5. Immune Responses in TDLNs and TME in the B16-OVA Tumor Model

Considering that the immunosuppressive tumor microenvironment (TME) may hinder the anti-cancer efficacy of the tumor-infiltrating lymphocytes, we subsequently explored the underlying cause of the diminished anti-tumor effect of the frequent administrations, with the aim of determining the optimal strategy. The leading exhaustion marker involved in tumor immune surveillance was evaluated in the TME (Appendix A) and TDLNs (Appendix A). As shown in Figure 6, compared with non-treated mice, immunosuppression-related PD-1^+^ CD8^+^ T cells were observed, with decreases of 540% and 340% in TDLNs (Figure 7a,b) and the TME (Figure 7c,d) after two vaccinations, respectively, indicating a potent anti-tumor efficacy. Nevertheless, PD-1^+^ CD8^+^ T cells’ proportion was evidently elevated after further boosting administrations of APE/OVA. Compared with APE-2, APE-4 increased the intratumoral PD-1^+^ CD8^+^ T cells by 220% in TDLNs and 150% in the TME. This indicated that the less potent anti-tumor efficacy of APE-4 may be attributed to the sudden increase in PD-1^+^ CD8^+^ T cell populations.

Lymphocyte activation gene-3 (LAG-3), an inhibitory immune checkpoint molecule comparable to PD-1, was further evaluated, as shown in Figure 8. Although APE down-regulated the expressions of LAG-3, higher frequencies of LAG-3^+^ CD8^+^ T cells were observed with further boosting doses. APE-4 allowed the proportion of LAG-3^+^ CD8^+^ T cells to increase by ~1-fold in both TDLNs and the TME compared to APE-2 (Figure 7a–d). These results suggested that dose-boosting at later times may induce the engagement of the PD-1^+^ CD8^+^ T cells and LAG-3^+^ CD8^+^ T cells, triggering the immune-regulatory effects.

### 3.6. Anti-Tumor Therapeutic Efficacy of APE Combined with Anti-PD-1 Therapy

It is worth noting that the binding between programmed cell death protein 1 (PD-1) and programmed cell death ligand 1 (PD-L1) participate in the process of immune escape during the development and metastasis of tumors. Blocking the immunosuppression-associated protein binding of PD-1 and PD-L1 has been proven to slow progression of a tumor’s expansion through regaining suppressed anti-tumor immunity. As aforementioned, frequent dosing of APE cancer vaccines may increase the engagement of PD-1^+^ CD8^+^ T cells. We assumed that the co-administration of APE with anti-PD-1 antibody may lessen the immunosuppressive effect and induce higher anti-tumor responses. As shown in Figure 9a, combined treatment of anti-PD-1 was given three times at D + 11, D + 14, and D + 17 for each group. In conjunction with anti-PD-1, with a stable body weight (Appendix A), APE-3 and APE-4 were effective in suppressing tumor growth in the B16-OVA subcutaneous tumor model and enhancing mouse survival, while APE-2 showed a suboptimal tumor regression and survival (Figure 9b,c). Accordingly, the immunosuppressive TME may be weakened by this cocktail therapy, allowing for the increased cellular immune responses after booster vaccinations.

## 4. Discussion

The scientifically rigorous standards of safety in clinical vaccines mean alum remains a crucial important benchmark of all adjuvants, effectively augmenting antibody titers and Th2 humoral immunity for protection, yet seldom inducing Th1 or Th17 cellular immunity [25], which are considered more important for vaccines against diseases such as malaria, tuberculosis, influenza, pertussis, and tumors [26]. Several studies have suggested the potential benefits of nanoscale alum adjuvants, which tend to induce stronger and more persistent antigen-specific IgG levels and augmented immunity, particularly T cell responses, compared to micron-sized aluminum hydroxide particles. However, the small size and large surface area of particles may lead to aggregation and finally affect the immune response [27]. Toll-like receptors (TLRs), which are localized at the cell membrane or in endosomal compartments, can be activated via conjugation with their agonists [28]. Numerous efforts have managed to combine TLRs agonists with aluminum to elicit a T cell response against aggressive tumors. The combination of CpG oligonucleotides (ODNs), which bind to TLR9, and alum adjuvant elicited antigen-specific antibody production and induced an IFN-γ-secreting T cell response. For CpG–alum, CpG tightly adsorbs to alum via electrostatic interactions that may affect the adsorptive capacity and strength of the antigens. Here, we adsorbed alum on the oil/water interphase to prepare alum-stabilized *Pickering emulsion* (APE). Besides the electrostatic and coordinate interactions with the alum, the alum-packed oil/water interphase also offered a larger specific surface and more hydrophobic interactions for large quantities of antigens and CpGs with multiple driving forces, which may have diminished the possibility of competitive adsorption. Furthermore, with the aid of the alum-stabilized droplets, antigens or CpG may have the potential to remain at the injection site and cause a robust and durable APC activation at the injection site, which is crucial for T cell activation. Meanwhile, with the positively charged surface, APE may exhibit antigen cross-presentation via lysosomal escape for the Th1-mediated immune response. FDA-approved squalene and alum may confer biosafety to APE after repeat immunizations.

Booster vaccinations have been frequently used to maximize anti-tumor efficacy and have been applied in most cancer vaccine studies [29]. However, incomplete Freund’s adjuvant (IFA), in a clinical trial of a melanoma vaccine, demonstrated that 1–3 injections induce accumulation of mature dendritic cells, while 4–6 injections induce FoxP3+ T cells and eosinophils and negative immune regulatory processes in the site [30]. This was in accordance with other recent studies, suggesting that the relative prevalence of T suppressor cells may be responsible for reducing lymphocyte activation and infiltration, causing the suboptimal anti-tumor effect. Here, we put extra emphasis on the connection between frequent vaccinations and the immune efficacy against tumors, and we obtained a similar result. Although more administrations of APE/OVA (APE-3 and APE-4) increased CD8+ T cell activations, a negligible difference in the anti-tumor potency was observed. After the prompted in-depth analysis of immune cells between the tumor microenvironment and tumor-draining lymph nodes, we found that the number of PD-1+ CD8+ T cells and LAG-3+ CD8+ T increased, leading to the immunosuppressive microenvironment to diminish the onset of the anti-tumor effect after the further booster injections of APE/OVA.

This indicated that repetitive antigenic stimulations may up-regulate the expression of negative regulatory markers (e.g., PD-1, LAG-3, TIM-3) among the tumor-infiltrating lymphocytes, leading to an elevated immune suppressive microenvironment [31]. Furthermore, the combined injection of anti-PD-1 antibody down-regulated the immune tolerance by binding to PD-1 receptor, bringing back the revitalized T cells. As shown in Figure 9, evident tumor regression and higher survival rates were detected after four APE/OVA vaccinations, suggesting that the relative prevalence of T suppressor cells may be responsible for reducing lymphocyte activation and infiltration, causing a suboptimal anti-tumor effect [32].

Accordingly, to further enhance the anti-tumor efficacy, we employed anti-PD-1 antibody, which can bind to PD-1 receptor to bring back the revitalized T cells. After four APE/OVA vaccinations, suppressed tumor growth was detected in the B16-OVA subcutaneous tumor model and the mouse survival was increased.

Taken together, APE may serve as a potent adjuvant for cancer vaccines. Moreover, stemming from the above data on APE vaccinations, the tumor regression caused by the cancer vaccines may not be increased by simply increasing the frequency of administrations. Elevated immunocompromised T cells and the microenvironment within the tumor may have major implications for hampered efficacy. Combining cancer vaccines with anti-PD-1 therapy may have greater potential in terms of the anti-tumor effect.

## Figures and Tables

**Figure 1 vaccines-11-01169-f001:**
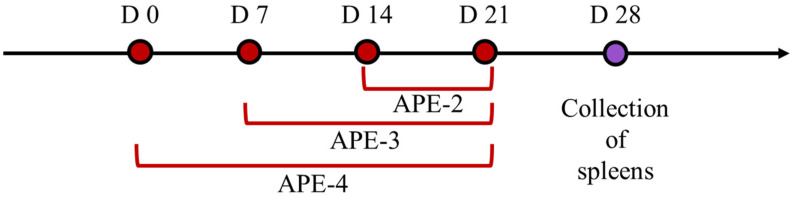
Schematic diagram of vaccination regimens. (Red dots: vaccinations of APE on different time points; Purple dot: collection and analyses of splenocyte on day 28).

**Figure 2 vaccines-11-01169-f002:**
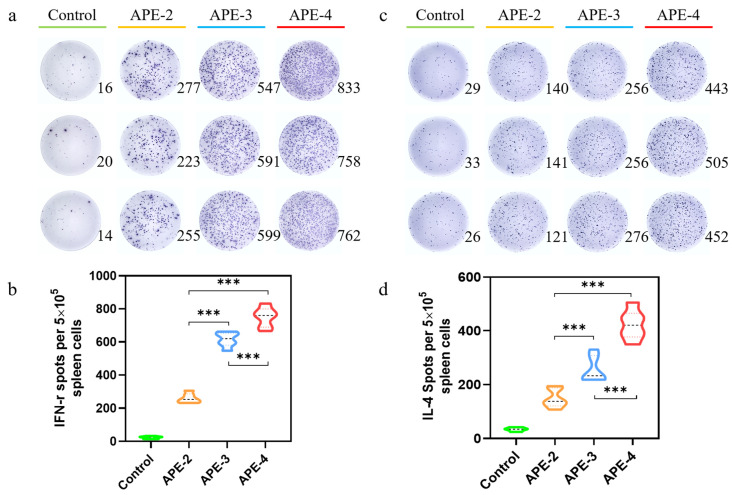
Potent adaptive response to APE/OVA tumor vaccinations. (**a**) ELISpot images of IFN-γ spot-forming cells among splenocytes. (**b**) Statistical analysis of IFN-γ spot-forming cells among the splenocytes. (**c**) ELISpot analysis of IL-4 spot-forming cells among splenocytes. (**d**) Statistical analysis of IL-4 spot-forming cells among the splenocytes. Data in the graphs are shown as arithmetic means ± s.e.m. from three independent experiments. *** *p*  <  0.001.

**Figure 3 vaccines-11-01169-f003:**
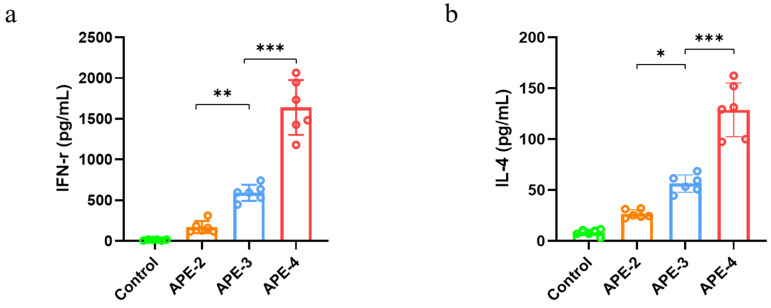
Secretion of cytokines. (**a**) IFN-γ and (**b**) IL-4 cytokine level in the splenocyte supernatant. The data are shown as the mean ± s.e.m. of a representative experiment (*n* = 6), * *p* < 0.05, ** *p* < 0.01, *** *p* < 0.001, analyzed by one-way ANOVA.

**Figure 4 vaccines-11-01169-f004:**
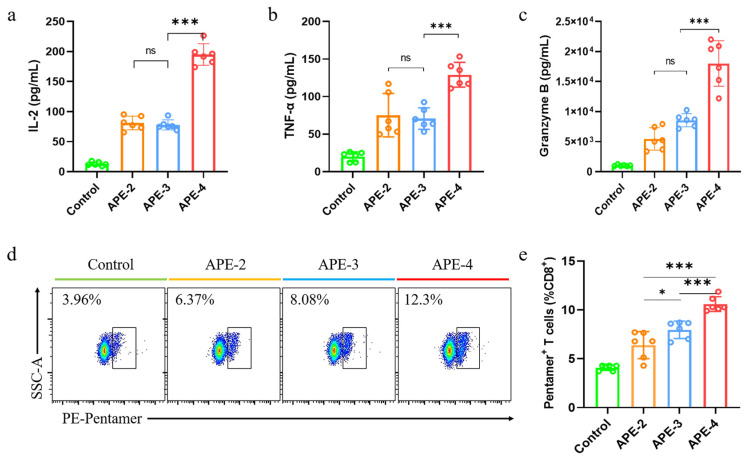
Secretion of cytokines. (**a**) IL-2 cytokine, (**b**) TNF-α cytokine secretion, and (**c**) granzyme B cytokine secretions among splenocytes. (**d**) Scatter plots and (**e**) statistical analysis of OVA-specific CD8^+^ T cells in the spleen. The data are expressed as the mean ± s.e.m. from a representative experiment (*n* = 6), ns, not significant, * *p* < 0.05, *** *p* < 0.001, analyzed by one-way ANOVA.

**Figure 5 vaccines-11-01169-f005:**
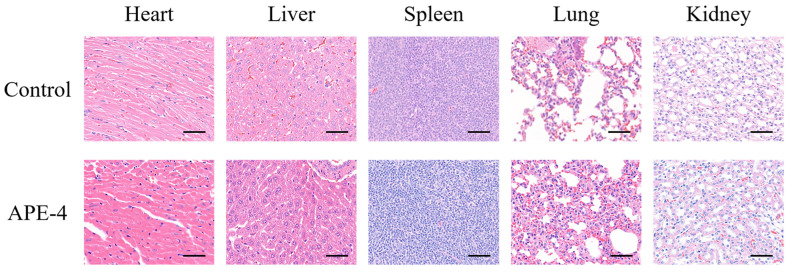
Biocompatibility evaluations via hematoxylin and eosin (H&E) staining of the key organ sections from C57BL/6 mice after four administrations (APE-4). Scale bar = 50 μm.

**Figure 6 vaccines-11-01169-f006:**
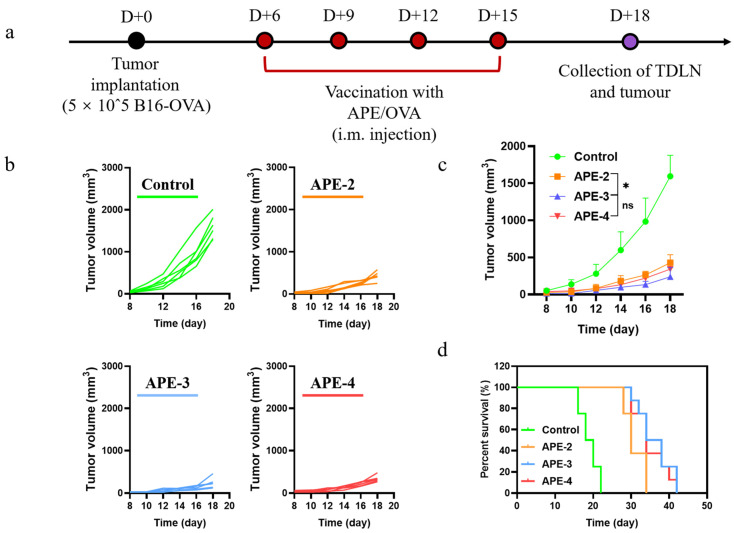
In vivo anti-tumor immune responses to APE/OVA in a B16-OVA tumor model. (**a**) Schedule of vaccinations in the B16-OVA tumor model. (**b**) Individual tumor growth kinetics of the B16-OVA tumors after different treatments. APE-2: two vaccinations on D + 6 and D + 9; APE-3: three vaccinations on D + 6, D + 9, and D + 12; APE-4: four vaccinations on D + 6, D + 9, D + 12, and D + 15. (**c**) Average tumor growth curves (*n* = 6). (**d**) Survival curves (*n* = 8) in the B16-OVA tumor model. The data are shown as the mean ± s.e.m. from a representative experiment (*n* = 6). ns, not significant, * *p* < 0.05, analyzed by two-way ANOVA.

**Figure 7 vaccines-11-01169-f007:**
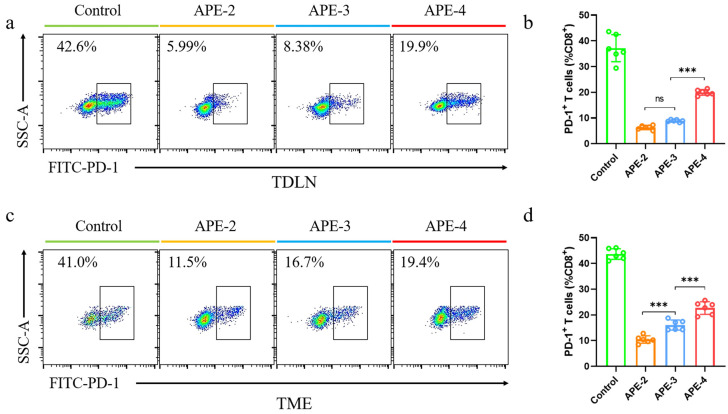
Immune cell analysis in a B16-OVA tumor model. (**a**) Representative flow cytometry dot plots and (**b**) percentage of PD-1^+^ cells in CD8^+^ T cells (CD3^+^ CD8^+^) in TDLNs. (**c**) Representative flow cytometry dot plots and (**d**) percentage of PD-1^+^ cells in CD8^+^ T cells (CD3^+^ CD8^+^) in TME. These data are shown as the mean ± s.e.m. from a representative experiment (*n* = 6), ns, not significant, *** *p* < 0.005, analyzed by one-way ANOVA.

**Figure 8 vaccines-11-01169-f008:**
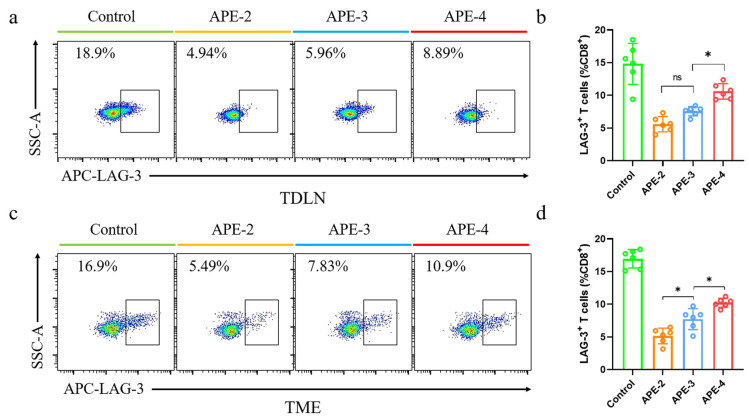
Immune cell analysis in a B16-OVA tumor model. (**a**) Representative flow cytometry dot plots and (**b**) percentage of LAG-3^+^ cells in CD8^+^ T cells (CD3^+^ CD8^+^) in TDLNs. (**c**) Representative flow cytometry dot plots and (**d**) percentage of LAG-3^+^ cells in CD8^+^ T cells (CD3^+^ CD8^+^) in TME. These data are shown as the mean ± s.e.m. from a representative experiment (*n* = 6), ns, not significant, * *p* < 0.05, analyzed by one-way ANOVA.

**Figure 9 vaccines-11-01169-f009:**
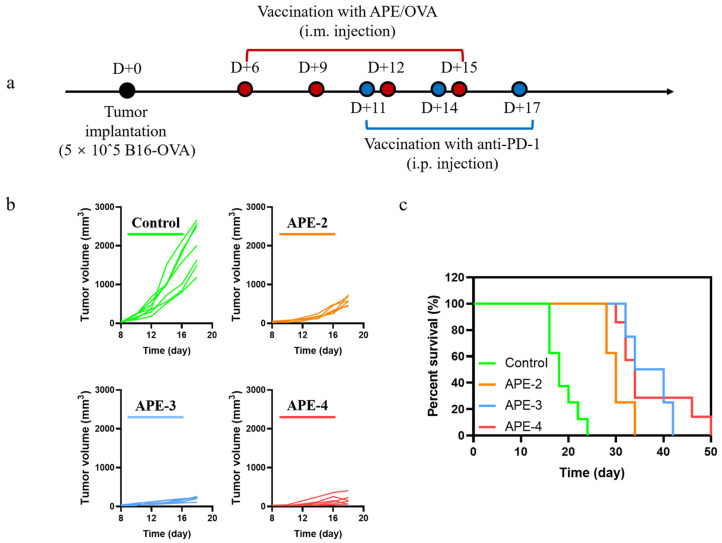
In vivo anti-tumor immune responses by APE/OVA combined with anti-PD-1 in a B16-OVA tumor model. (**a**) Schedule of vaccinations in the B16-OVA tumor model. (**b**) Individual tumor growth kinetics of the B16-OVA tumors after different treatments. APE-2: two vaccinations on D + 6 and D + 9 combined with three treatments of anti-PD-1; APE-3: three vaccinations on D + 6, D + 9, and D + 12 combined with three treatments of anti-PD-1; APE-4: four vaccinations on D + 6, D + 9, D + 12, and D + 15 combined with three treatments of anti-PD-1. (**c**) Survival curves (*n* = 8) in the B16-OVA tumor model. The data are shown as the mean ± s.e.m. from a representative experiment (*n* = 6).

**Table 1 vaccines-11-01169-t001:** The characterization of alum and APE.

	Size (nm)	Zeta Potential (mV)	OVA Loading Efficiency ^1^ (%)
Alum	847.0 ± 18.9	+17.1 ± 0.3	97.9 ± 0.4
APE	2761.0 ± 179.4	+30.6 ± 2.4	98.1 ± 0.6

^1^ The OVA antigen loading efficiency was calculated by the proportion of adsorbed OVA and the total amount of the added OVA. OVA antigens were mixed with the APE or alum for 1 h at 25 °C. Data are shown as the mean ± s.d. (*n* = 3).

## Data Availability

Not applicable.

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
