# Peer review of "Development and Optimal Immune Strategy of an Alum-Stabilized Pickering emulsion for Cancer Vaccines"

_vaccines, 2023, doi:10.3390/vaccines11071169_

Round 1
Reviewer 1 Report
The manuscript titled " Development and optimal immune strategy of alum-stabilized Pickering emulsion for cancer vaccines" is a good work wherein the authors have reported synthesis of an Alum-stabilized Pickering Emulsion (APE) that can load a high number of antigens which continue to release for extensive maturation and activations of antigen presenting cells (APCs). However I have detected certain issues which need to be addressed extensively for improving the quality of the paper.
1. In the introduction section, the authors are not very clear about the need to carry out the research.
2. Many grammatical errors and English language issues mar the quality of the paper.
3. I could find many literature precedents with almost the same results. The authors should clearly mention all related previous published works and highlight the improvements made in their work.
Many grammatical errors and English language issues mar the quality of the paper. Extensive language and grammatical corrections need to be carried out.
Author Response
We deeply appreciate and treasure the reviewer’s comments and the constructive suggestions, which strongly inspired and encouraged us to work on the revised manuscript. In light of the comments, we have carefully collected, analyzed and summarized the relative literature and made some revisions, and the details have been put in the attachment, please check it.

Reviewer 2 Report
This paper describes experimental observations in relation to the efficay of an adjuvant for cancer vaccination. It extends previous significant work developed by the same authors regarding the fine tunning of the adjuvant.
The paper is well written and understandable. Methods are well described. Figures are well presented
The English text should be carefully revised and improved.
While results support the efficacy of the vaccination procedure, the Discussion of this paper is lacking a comparison with other applied adjuvants and vaccination protocols, in such a way that the authors should highlight the advantages and limitations of their methodology.

The English text should be carefully revised and improved.
Author Response

(The authors gave the same response as above.)

Reviewer 3 Report
The authors of the paper “Development and optimal immune strategy of alum-stabilized Pickering emulsion for cancer vaccines” give us an overview of a potential usage in cancer vaccines of a stabilized alum-squalene adjuvant.
The article need English editing, but major and minor issues are found in this article:
Major issues:
1- Alum-stabilized Pickering emulsion: There is no evidence of a chemical/biological stability of this adjuvant.
2- Method part: The method part should be re-written. The vaccine schedule is missing. and we do not understand the downstream analysis.
3- Results and Discussions: There is no discussion in the article. References are missing in this crucial part of the article.
4- Supplemental figures are also missing in the submission, we cannot review them.
5- In this study, we miss the important comparison with Alum-OVA only (not as APE). This is a crucial result to see the real biological impact of this new formulation.
6- Figure 5d and Figure 8c: The Vaccination and the Vaccination+anti-PD1 protocol are finally not protective, even with 3 and 4 immunization. We do not see any greater potential for potent anti-tumor effect.
Minor issues:
1- Line 102: 2.3 Cell culture, this is not the real title as you are talking about mice.
2- Line 157, why you resuspend cells in whole blood and tissue diluent?
3- 2.9 and 2.10, did you evaluate the weight loss of your mice during the protocol? This is important to check the illness of these mice.
4- Table 1, why you have this enormous size difference between Alum and APE? They both have similar loading efficiencies.
5- Line 214, why you talk about COVID19? No reference.
6- Figure 4: Pictures are very small; we cannot see if there are immune cell infiltration differences.
The article need moderate English editing.
Author Response
We deeply appreciate and treasure the reviewer’s comments and the constructive suggestions, which strongly inspired and encouraged us to work on the revised manuscript. In light of the comments, we have carefully collected, analyzed and summarized the relative literature and made some revisions, and the details have been put in the attachment. Please check it.

Round 2
Reviewer 2 Report
The authors have paid attention to former comments, and have greatly modified and improved the Introduction and Discussion of the paper, providing a convincing answer to those. Accordingly, References have also been revised.
Maybe the Section 3 should better read just “Results” instead of “Results and Discussions”.
Section 4 might be “Discussion” or “Discussion and conclusions”.
The English text contains some mistakes that should be corrected.

The English text contains some mistakes that should be corrected.
Author Response
We are encouraged by the reviewer’s comment. We also want to express our gratitude for the referee’s dedication and support for our work, which aided in preparing the revised manuscript. We apologize for the mistakes in the writings of the previous version. In light of the comment, The Section 3 and the Section 4 have been revised to “Results” and“Discussion” respectively. Then, we have carefully checked and improved the use of English, which have been highlighted in yellow in the revised manuscript. We sincerely appreciate reviewer’s work, and hope that it would meet with the expectation.

Reviewer 3 Report
Thank you for editing the manuscripts regarding my comments.
Better English than the previous version.
Author Response
We are encouraged by the reviewer’s comment. We also want to express our gratitude for the referee’s dedication and support for our work, which aided in preparing the revised manuscript. We apologize for the mistakes in the writings of the previous version. In light of the comment, we have carefully checked and improved the use of English, which have been highlighted in yellow in the revised manuscript. We sincerely appreciate reviewer’s work, and hope that it would meet with the expectation.
